# Genome-Wide Identification, Evolution and Expression of the Complete Set of Cytoplasmic Ribosomal Protein Genes in Nile Tilapia

**DOI:** 10.3390/ijms21041230

**Published:** 2020-02-12

**Authors:** Gangqiao Kuang, Wenjing Tao, Shuqing Zheng, Xiaoshuang Wang, Deshou Wang

**Affiliations:** Key Laboratory of Freshwater Fish Reproduction and Development (Ministry of Education), Key Laboratory of Aquatic Science of Chongqing, School of Life Sciences, Southwest University, Chongqing 400715, China; kuanggangqiao@126.com (G.K.); enderwin@163.com (W.T.); zhengsq0825@163.com (S.Z.); wxs958386@163.com (X.W.)

**Keywords:** cytoplasmic ribosomal protein gene, evolution, transcriptome analysis, Nile tilapia

## Abstract

Ribosomal proteins (RPs) are indispensable in ribosome biogenesis and protein synthesis, and play a crucial role in diverse developmental processes. In the present study, we carried out a comprehensive analysis of RPs in chordates and examined the expression profiles of the complete set of 92 cytoplasmic RP genes in Nile tilapia. The RP genes were randomly distributed throughout the tilapia genome. Phylogenetic and syntenic analyses revealed the existence of duplicated RP genes from 2R (RPL3, RPL7, RPL22 and RPS27) and 3R (RPL5, RPL19, RPL22, RPL41, RPLP2, RPS17, RPS19 and RPS27) in tilapia and even more from 4R in common carp and Atlantic salmon. The RP genes were found to be expressed in all tissues examined, but their expression levels differed among different tissues. Gonadal transcriptome analysis revealed that almost all RP genes were highly expressed, and their expression levels were highly variable between ovaries and testes at different developmental stages in tilapia. No sex- and stage-specific RP genes were found. Eleven RP genes displayed sexually dimorphic expression with nine higher in XY gonad and two higher in XX gonad at all stages examined, which were proved to be phenotypic sex dependent. Quantitative real-time PCR and immunohistochemistry ofRPL5b and RPL24 were performed to validate the transcriptome data. The genomic resources and expression data obtained in this study will contribute to a better understanding of RPs evolution and functions in chordates.

## 1. Introduction

Ribosomes are absolutely essential for life, generating all proteins required for cell growth and maintenance. In eukaryotes, a mature ribosome consists of four different ribosomal RNAs (rRNAs; 18S in the small subunit and 28S, 5.8S, and 5S in the large subunit), as well as ~80 ribosomal proteins (RPs) [1]. RPs from the small and large subunits are named as RPS and RPL, respectively. Great progress has been made in recent years in elucidating the structure and function of the RPs [2,3]. The cellular levels of some RP transcripts change as a function of growth, development and certain tumors [4,5,6]. In addition, many RPs are believed to have important functions in various other cellular processes, the so-called extraribosomal functions [7,8]. Over the past years, mutations in ribosomal proteins or ribosomal biogenesis factors have been identified in patients with varying disease [9,10]. For example, haploinsufficiency of the RPS4 genes has been suggested to contribute to anatomic abnormalities associated with the Turner syndrome in humans [11]. In zebrafish, many ribosomal protein mutations are associated with growth impairment and tumor predisposition [12]. Knockdown of RPS7 causes developmental abnormalities via p53 dependent and independent pathways [13].

Genomic mapping of the RPs is of great interest because of its value in understanding genome evolution. The sequence information of these genes is required for mapping them onto chromosomes. The large number of RPs has complicated the quest for a complete understanding of their sequence information, gene structures and genomic organizations. The first complete identification of human ribosomal proteins was published in 2001 [14,15]. Although fish represent the largest group of vertebrates, a complete map of RP genes from any teleost species has not been available until recently. The complete set of RP genes has been identified from cDNA library in channel catfish, Senegalese sole and Atlantic halibut [16,17,18,19]. However, the exact number of RP genes in fish species is unclear due to lack of complete genome sequences. The rapid development of genomics, sequencing and disclosure of more and more animal genomes in recent years has made it feasible to elucidate the evolution of RP genes in vertebrates.

The expression pattern of RP genes in different tissues and developmental stages has been reported in several species. In *Haemaphysalis longicornis*, RPS-27 was expressed in all life stages, and it was more highly expressed in the salivary gland than in the midgut at the tissue level [20]. In channel catfish, Senegalese sole and Atlantic halibut, RP genes were expressed ubiquitously at a similar level among tissues, although they differed in the relative abundance of their transcripts [16,17,18,19]. Sexually dimorphic expression of RP genes was also observed in some species. For example, two ribosomal proteins, RPL17 and RPL37, were expressed higher in males compared to females in the zebra finch brain [21,22,23]. Later, RPS6 was also found to be enhanced in the male zebra finch brain as compared to the female’s throughout life [24]. In marine shrimp, RPL24 was differentially expressed in ovary and testis [25]. In zebrafish, RPL39 and RPL23A were found to be the female-biased genes [26]. Existing studies on global RP gene expression are mainly based on expressed sequence tag (EST) approach, the expression level of genes probably not represented properly. The advances of transcriptomics make it possible to carry out more accurate expression analyses of the complete set of RPs in different tissues and different developmental stages in vertebrates.

The Nile tilapia (*Oreochromis niloticus*) is a commercially important farmed fish species in aquaculture worldwide, with the growth rate of males substantially higher than that of females [27]. Because of the key role RPs play in cellular growth and proliferation, it is important to elucidate the expression pattern of RPs in different tissues in tilapia. As is known to all, during fish gonadal differentiation and development, the male and female gonads display different gene expression profiles. Thus, the gonadal model provides a unique opportunity to delineate the coordinated expression of RPs and other genes during gonadal differentiation and development. However, the complete set of Nile tilapia ribosomal protein genes has not been previously documented or characterized, and the expression of RP genes during gonadal differentiation and development in fish has not been reported to date. The availability of the whole genome sequences of tilapia and tissue transcriptomes [28,29], together with gonadal transcriptomes at different developmental stages [30,31], has made it an excellent model for genome-wide identification, tissue distribution and gonadal expression profile investigation of RP genes involved in gonadal differentiation and development.

RPs are highly conserved even between vertebrates and invertebrates. This sequence identity enables us to name RPs with significant confidence and also enables the use of antibodies raised against mouse RPs on tilapia proteins. In the present study, we carried out a comprehensive analysis of RP genes in tilapia and other chordates, including genome-wide search, chromosomal location, phylogeny, synteny and spatial and temporal expression profiles. Our results provided a new insight into evolution of RPs and revealed their potential role in teleosts.

## 2. Results

### 2.1. Identification of RP Genes in Tilapia

To obtain the complete set of RP genes, a blast search against tilapia genome sequences at NCBI was performed using zebrafish and human RP genes as reference. A total of 92 RP genes, including 55 encoding large-subunit proteins and 37 encoding small-subunit proteins (Table 1), were identified in tilapia. The GenBank accession number, Chromosome location, number of exons, gene length (bp) and protein length (aa) for the complete set of RP genes are shown in Table 1. All the 92 RP genes have complete open reading frames (ORFs) and gene sequences. The average size of the genes from the transcription start site is ~3.61 kb. RPS17a is the largest (~7.98 kb), whereas RPL41bis the smallest (only ~0.91 kb). Each gene has an average of 5.5 exons, ranging from 3 (RPL39 and RPS29) to 10 (RPL3-2 and RPL4). All of the 92 RP genes were assigned to the linkage maps of the tilapia genome (Figure 1). The RP genes were found to be randomly distributed throughout the tilapia genome covering all 22 linkage groups (LG). Each LG carried at least one RP gene. Both the LG7 and LG22 carried a total of nine genes. LG19 is the only example with only one RP gene (RPL13A) present.

### 2.2. Comparative Analysis of RP Genes in Chordates

To understand the conservation of RP genes in vertebrates, the putative amino acid sequences of tilapia RPs were compared with those from the zebrafish, medaka, fugu and human (Appendix A). Overall identities of tilapia RPs were97%, 96.5%, 94.1%and 92.7% to those of medaka, fugu, zebrafish and human, respectively. The most conserved RPs in the five species (overall mean identity ≥98%) were RPL23, RPL38, RPL39, RPL40, RPS13, RPS14, RPS23 and RPS27A.

The numbers of amino acids in the RPs are highly conserved among species. Among the five species compared, 48 of the 92 RPs (52.2%) have the same number of amino acids. The other RPs display different sizes in either one species (RPL5a, RPL9, RPL18 and RPL28 in zebrafish and RPL10A, RPL13A, RPL23A, RPL29, RPS3, RPS10, RPS27-1a and RPS19a in human), or two to four species (21 RPs including RPL4, RPL6, RPL7-1, RPL10, RPL14, RPL19a, RPL19b, RPL22-1b, RPL22-2, RPL29, RPL30, RPL31, RPLP0, RPLP1, RPLP2a, RPSA, RPS2, RPS3A, RPS11, RPS24 and RPS25) or even five species (RPL4, RPL14 and RPL29). In all three cases, fish RPs have fewer amino acids than their mammalian counterparts.

Given that the whole genome duplication (WGD) can drive the expansion of gene families, we surveyed the number changes of the RP genes in 13 representative chordates with different rounds of WGD, from first round to fourth round (referred to as 1R, 2R, 3R and 4R) (Figure 2, Figure 3 and Appendix A). After comprehensive analysis, we identified 79, 64, 86, 84, 83, 80, 85, 125, 90, 148, 88, 92 and 86 RP genes in sea squirt, lamprey, coelacanth, human, mouse, spotted gar, zebrafish, common carp, channel catfish, Atlantic salmon, medaka, tilapia and fugu genome, respectively. Compared with Atlantic salmon, the absence of certain RPs in common carp may be due to more secondary loss or incomplete genome sequencing and assembly.

### 2.3. Phylogenetic and Syntenic Analyses of RP Paralogous Genes in Tilapia

Phylogenetic and syntenic analyses were performed to understand the evolution of RP genes in vertebrates. Ten RP genes with different numbers of paralogous genes were identified in tilapia, including two for RPL3, RPL5, RPL7, RPL19, RPL41, RPLP2, RPS17 and RPS19, three for RPL22, and four for RPS27. All of these RP paralogous genes were unevenly distributed throughout the genome as shown in Figure 1. Interestingly, RPS19a and RPS27-1a, RPS19b and RPS27-1b, were located on LG11 and LG22, respectively.RPL41a and RPL22-1a, RPL41b and RPL22-1b, were located on LG20 and LG5, respectively. In contrast, both RPL3-1 and RPL3-2 were located on LG4.

A phylogenetic analysis was carried out for these duplicated genes, including sequences available in the GenBank and Ensembl for representative fish and tetrapod counterparts (Appendix A). Paralogs of RPL41 were not included in this analysis due to their small ORFlength and identical amino acid sequence (25 aa). In the tree, paralogs of RPL3-1 and RPL3-2 were grouped into two distinct well-supported clades that included the fish and tetrapod counterparts (Appendix A). Similarly, the RPL7-1 and RPL7-2 paralogs also clustered into two separate sister clades (Appendix A). The phylogenetic relationship of these genes suggests that they could have arisen from 2R and evolved independently. Paralogs of RPL22-1a/RPL22-1b and RPL22-2 were grouped into two paraphyletic groups, including fish and tetrapod counterparts as shown in Figure 3A. RPL22-1b appeared closely related to the other fish RPL22 sequences, although they did not form a well-resolved clade (Appendix A). The four paralogous genes for RPS27 in tilapia clustered into two paraphyletic groups with their fish and tetrapod counterparts (Appendix A). The tilapia RPS27-1a and RPS27-1b showed a monophyletic origin with tetrapod counterparts as a paraphyletic group, whereas tilapiaRPS27-2b grouped with other fish counterparts in a well-supported clade, and tilapiaRPS27-2awas closely related to its cichlid counterparts and RPS27-2 sequences from tetrapod. Both theRPL5 and RPL19 paralogs showed a monophyletic origin with tetrapod counterparts as a paraphyletic group (Appendix A, respectively). RPLP2 paralogs clustered into two separate sister clades (Appendix A). RPLP2b in tilapia grouped with other fish counterparts in a well-supported clade, whereas the RPLP2a in tilapia was grouped with counterparts from medaka, fugu, three-spined stickleback, Atlantic salmon, channel catfish, zebrafish, carp and RPLP2 from spotted gar, coelacanth and tetrapods. Paralogs of RPS17 also clustered into two separate sister clades (Appendix A). RPS17b in tilapia grouped with other fish counterparts in a well-supported clade, whereas the RPS17a was closely related to counterparts from medaka, three-spined stickleback, fugu, zebrafish, carpand RPS17 sequences from spotted gar, coelacanth and tetrapod.Two RPS19paralogs were found in tilapia for the first time(Appendix A), and they share 84.2% identity at the amino acid level. Similar to RPL5 and RPL19, the RPS19 paralogs also showed a monophyletic origin with tetrapod counterparts as a paraphyletic group, but the RPS19b was only detected in species from Percomorpha, including Perciformes, Cyprinodontiformes and Pleuronectiformes.

In addition, synteny analysis showed that all the duplicates of tilapia RP genes and their adjacent genes were in regions of conserved synteny in teleosts and other vertebrates (Figure 4 and Appendix A). Phylogenetic and syntenic analyses revealed the existence of paralogous genes from 2R (RPL3, RPL7, RPL22 and RPS27) and 3R (RPL5, RPL19, RPL22, RPL41, RPLP2, RPS17, RPS19 and RPS27). RPL3 and RPL7 only experienced 2R event, and RPL22 and RPS27 experienced both 2R and 3R events. In addition, duplication ofRPS19 was observed only in Percomorpha. Duplicates of RPL5, RPL19, RPL41, RPLP2, RPS17 and RPS19were clustered in two independent clades in the teleost lineage, whereas only one copy was found in one clade in tetrapods and spotted gar. These results indicated that those duplicates were derived from 3R event after teleost fish diverged from spotted gar.

### 2.4. Tissue Distribution and Ontogeny Expression of RP Genes in Gonads of Tilapia by Transcriptomic Analysis

Transcriptome data from eight adult tissues and gonads from four developmental stages of tilapia were analyzed to understand the expression profile of RP genes. The RP genes were found to be expressed in all tissues examined (Figure 5). Interestingly, most RP genes exhibited tissue-biased expression patterns especially in heart, liver, muscle and head kidney. RPS10, RPS12, RPS14, RPS18, RPS26, RPL37 and RPL38 were highly expressed, while RPL3-2 showed background expression level in all eight tissues. In addition, RPL38 was the most highly expressed in all eight tissues.

Most RP genes were generally highly expressed in tilapia gonads (Figure 6). The total and average reads per kb per million (RPKM) values were 804,840 and 8748 for XX gonads, and 866,548 and 9419 for XY gonads, respectively (Table 2). No sex- and stage-specific RP genes were observed. With regard to single RP genes in the four developmental stages, 57 and 69 RP genes were found to express at an average RPKM value above 1000 in XX and XY gonads, respectively. In contrast, some RP genes were rarely expressed both in XX and XY gonads at the four developmental stages, including RPL3-2, RPL7-2, RPS19b, RPS27-1b and RPS28, which were present at an average RPKM value lower than 100.

At 5 dah (day after hatching), the total and average RPKM valuesof the RP genes were 409,026 and 4446 for XX gonads, and 229,345 and 2493 for XY gonads, respectively (Table 2). Twenty-eight and 25 RP genes were expressed at RPKM values above 1000 in XX and XY gonads, respectively. Thirty-two RP genes were expressed at 2-fold higher levels in XX than XY gonads (XY/XX RPKM value < 0.5), and 10 genes showed an approximately 2-fold higher expression in XY than XX gonads (XY/XX RPKM value > 2), 50 genes displayed no significant differences in expression level between XX and XY gonads (XY/XX RPKM value = 0.5–2) (Figure 6 and Figure 7).

At 30 dah, the expression levels of RP genes in XX gonads were down-regulated with the total and average RPKM values dropping to 299,234 and 3253, respectively, while in XY gonads they were up-regulated with the total and average RPKM values rising to 372,252 and 4046, respectively (Table 2). Seventy-one and 72 genes were expressed at RPKM values above 1000 in XX and XY gonads, respectively. At this stage, all RP genes except one (RPS19b with RPKM values of 1.47 and 3.76 in XX and XY gonads, respectively) displayed no significant differences in expression level between XX and XY gonads (XY/XX RPKM value= 0.5–2) (Figure 6 and Figure 7).

At 90 dah, the expression levels of RP genes were remarkably down-regulated in both XX and XY gonads, with the total and average RPKM values decreased to 45,903 and 499 for XX and 133,854 and 1455 for XY, respectively (Table 2). Six and 55 RP genes were expressed at RPKM values above 1000 in XX and XY gonads, respectively. Eighty-one out of 92 genes (88%) were expressed 2-fold higher in XY than XX gonads (XY/XX RPKM value > 2), and only 1 RP gene was expressed at 2-fold higher levels in XX than XY gonads (XY/XX RPKM value < 0.5), 10 genes displayed no significant differences in expression level between XX and XY gonads (XY/XX RPKM value = 0.5–2) (Figure 6 and Figure 7).

At 180 dah, the expression patterns of RP genes in both XX and XY gonads were similar to those at 90 dah, the total and average RPKM values of the RP genes were 50,678 and 551 for XX gonads, and 131,096 and 1425 for XY gonads, respectively (Table 2). Ten and 55 RP genes were expressed at RPKM values above 1000 in XX and XY gonads, respectively. Seventy out of 92 genes (76%) were expressed at 2-fold higher levels in XY than XX gonads (XY/XX RPKM value > 2), and only 1 RP gene was expressed at 2-fold higher levels in XX than XY gonads (XY/XX RPKM value < 0.5), 21 genes displayed no significant differences in expression level between XX and XY gonads (XY/XX RPKM value = 0.5–2) (Figure 6 and Figure 7).

Overall, the number and expression levels of differentially expressed RP genes were higher in XX than in XY gonads at 5 dah stage, while contrasting expression patterns were observed at 90 and 180 dah stages. At 30 dah stage, all RP genes except one (RPS19b) displayed no significant differences in expression level between XX and XY gonads. In addition, 11 RP genes displayed sexually dimorphic expressionin the gonad with 9 expressed higher in XY gonad and 2 higher in XX gonad at all stages examined. Transcriptome analysis of gonads from the sex-reversed fishrevealeda reversed expression profile of these 11 RP genes, indicating the sexual dimorphism is phenotypicsex dependent (Figure 8). Taken together, our results revealed a strong sex- and stage-dependent expression pattern of RP genes in tilapia gonad.

### 2.5. Validation by qRT-PCR and Immunohistochemistry

Immunohistochemistry (IHC) and qRT-PCR were performed to validate the transcriptome data. RPL5b and RPL24, the most significantly differentially expressed genes in XX and XY gonads and with commercial antibodies available, were selected for validation. By qRT-PCR, RPL5b was continuously expressed higher while RPL24 was continuously expressed lower in XX than XY gonads at 5, 10, 20, 50, 70 and 180 dah, even though their expression varied at different stages of gonad development (Figure 9).

By immunohistochemistry, strong specific signals of RPL5b were observed mainly in the cytoplasm of oocytes in the ovary, while weak signals were detected in the spermatocytes in the testis. However, nearly no signals were detected in other spermatogenic cells (Figure 10A,B). In contrast, RPL24 was found to be expressed ubiquitously at high levels in different spermatogeniccells, while very weak signals were observed in the cytoplasm of oocytes in the ovary (Figure 10C,D).

## 3. Discussion

### 3.1. Evolution of RP Genes in Chordates

In prokaryotic genomes, RP genes were found to be clustered in operons [32]. In the *Arabidopsis* genome, RP genes were reported not to be uniformly distributed with much higher density in several regions [33]. In rice, the RPS genes were found to be distributed throughout the rice genome. Both arms of the chromosome randomly carried the RPS genes. Each chromosome carried at least one member of the RPS gene family [34]. In humans, RP genes are widely scattered across the genome, both sex chromosomes and 20 autosomes (all but chromosomes 7 and 21) were found to carry one or more RP genes [15]. In the present study, the complete set of 92RP genes were randomly distributed throughout the tilapia genome, with each LG carrying one or more genes, similar to the RP gene distribution pattern observed in rice and humans.

Ribosomal proteins are indispensable in ribosome biogenesis and protein synthesis, and play a crucial role in diverse developmental processes. The rapid development of genome sequencing and bioinformatics has increased the availability of complete sets of RPs for a wide range of species, which allowed their application in phylogenetic analysis [35,36,37,38]. The identification and characterization of the RPs in channel catfish, Senegalese sole and Atlantic halibut add more molecular markers for studying genome evolution and phylogenetic relationships in teleosts [16,17,18,19]. However, the exact number of RP genes in chordates, especially in teleosts, has not been fully understood yet. In this work, we identified 79, 64, 86, 84, 83, 80, 85, 125, 90, 148, 88, 92 and 86 RP genes in the sea squirt, lamprey, coelacanth, human, mouse, spotted gar, zebrafish, common carp, channel catfish, Atlantic salmon, medaka, tilapia and fugu genome, respectively. We also updated the number of the RP genes from 80 [15] to 84in humans because of the isolation of four more paralogs of RPS27, RPL3, RPL7 and RPL22 from the human genome. These results revealed that the number of RP genes does not change much in chordates following 2R and 3R events, and significant expansion of RP genes is only observed in teleost fishes with 4R. This work should serve as a basis to allow comparative analysis of genome evolution and function of RP genes inchordates, especially in teleosts.

Four rounds of large-scale genome duplications (referred to as 1R, 2R, 3R and 4R) shaped genome evolution in fish [39,40]. Whole genome duplication events, followed by deletion or decay of some of the RP genes, are the major contributors to the diversity of models of evolution of RP genes. RP genes, which are highly expressed, exist in many copies and are essential for ribosomal function [41,42]. Different gene copies have been described for some RPs in fish. For example, channel catfish have two paralogous genes for RPL5, RPS26 and RPS27 [16,17]. In Senegalese sole, two paralogous genes for RPL13A, RPL19 and RPS27, as well as three different RPL22 genes have been identified. In Atlantic halibut, two paralogous genes for RPL3, RPL7, RPL19, RPL22, RPL41and RPLP2, and four different RPS27 genes were identified [18,19]. In the present study, paralogous genes for10RP types were identified in tilapia genome, of which RPL3, RPL5, RPL7, RPL19, RPL22, RPL41, RPLP2, RPS17 and RPS27 paralogs could be found in most fish species, including channel catfish, Senegalese sole and Atlantic halibut. We newly identified RPS19 paralogs in tilapia, which were also detected in other species from Percomorpha but not in other teleosts, indicating that they are derived from lineage-specific duplication.

RPL3 and RPL7 could have originated from 2R duplications since a two-branch tree topology containing the fish and tetrapod counterparts was observed. The 3R duplication would explain the RPL22-1 paralogs (RPL22-1a and RPL22-1b) observed in tilapia and other teleosts, which is closely phylogenetically related to RPL22-1 found in species without 3R and to RPL22-2 derived from 2R. RPL5, RPL19 and RPS17 obtained a paralogfrom3R duplication in teleosts [17,19], and one more for RPL5 in Atlantic salmon and rainbow trout with 4R. Although the previous study indicated the RPS27 gene as a mammalian-specific isoform [43], the identification of four RPS27 genes in tilapia supports the hypothesis of two RPS27 paralogs in tetrapod and at least two in fish as a common feature. The two additional paralogousRPS27genes might have appeared in the 3R or fish-specific genome duplication [16,18]. Both of the two paralogs in tetrapods seem to have duplicated in teleosts, making four RPS27 genes more likely where a few species have lost some of these four. This was further supported by the isolation of eight RPS27 genes in species with 4R. In summary, most of the duplicated RP genes from 2R were lost, but four of them, including paralogs of RPL3, RPL7, RPL22 and RPS27, were retained in vertebrates including human. Two of them (RPL22 and RPS27) even experienced 3R and retained 3 to 4 copies in teleosts. Taken together, our data clearly support the birth-and-death model for the evolution of RP genes.

### 3.2. Possible Roles of RP Genes in Different Tissues, Especially in Gonads

Spatial and temporal gene expression patterns are important for understanding gene regulation and function [44,45]. RPs are key components of the translational machinery responsible for protein synthesis in all cells, and thereby participate in multiple cellular processes including growth and development [46,47]. In zebrafish, RP genes were found to be expressed in a model of continuous coordinate increase from the onset of mid-blastula transition to hatching [6,48]. In Atlantic halibut, expression levels of 40 and 41 RPs were increased from embryos to 1-day-old yolk sac larvae and in fast skeletal muscle in juveniles, respectively [49]. In channel catfish, Senegalese sole and Atlantic halibut, the expression profiles of RP genes in tissues have been shown to be associated with the protein biosynthetic requirements and cellular demands [16,17,18,19]. In the present study, transcriptomic analysis revealed that the RP genes expressed in all tissues examined, but their expression levels differed among different tissues in tilapia, indicating their essential roles in various physiological processes.

To date, there are a few researches focused on the roles of RPs involved in gonad differentiation and development. In Chinese mitten crab, RPS27 and RPL40 were found to play key roles in gametogenesis and reproductive success [50]. In the present study, RP gene expression profile was obtained from transcriptome analyses of the gonad samples from tilapia at 5, 30, 90 and 180 dah, which represent sex determination and differentiation, initiation of germ cell meiosis in ovary, initiation of germ cell meiosis in testis, and vitellogenesis in ovary and sperm maturation in testis, respectively [30]. In tilapia gonads, most RP genes were highly expressed, and some of them displayed sexually dimorphic expression at different stages of development. At 5 dah, the number and expression levels of differentially expressed RP genes were higher in XX than XY gonads. At 30 dah, all RP genes except one displayed no significant differences between XX and XY gonads. At 90 and 180 dah, the number and expression levels of differentially expressed RP genes were higher in XY than XX gonads. The expression profile of RP genes in XX and XY gonads is similar to that of the steroidogenic enzyme genes and clearly associated with the biosynthesis of different numbers and types of proteins needed for gonadal differentiation and development, as revealed by gonadal transcriptomic analyses [30]. In addition, 11 RP genes displayed sexually dimorphic expression (phenotype dependent as revealed by analysis of the sex-reversed fish) in gonads with 9 (RPL7-1, RPL8, RPL11, RPL14, RPL18, RPL23A, RPL24, RPS5 and RPS24) expressed higher in XY gonad and 2 (RPL5b and RPS27-1b) higher in XX gonad at all stages examined. qRT-PCR and IHC of RPL5b and RPL24 validated the transcriptome data. The sexually dimorphic expression of these 11 RP genes at 5 dah indicates that they may play a crucial role in sex differentiation in tilapia. Furthermore, RPL7-1, RPL8, RPL11, RPL14, RPL18, RPL23A, RPL24, RPS5 and RPS24 may play important roles in the development of testis and spermatogenesis, while RPL5b and RPS27-1b may play important roles in the development of ovary and oogenesis. Sex-specific expression of translation elongation factors *eEF1α1b* in the testis and *42Sp50* in the ovary have been reported in frogs and several fish species, including tilapia [51,52,53,54]. Taken together, these results indicate that the gonad tightly controls its biosynthesis machinery to meet the needs of proteins required for oogenesis and spermatogenesis.

## 4. Materials and Methods

### 4.1. Animal Rearing

Nile tilapia fishes used in this study were reared in recirculating freshwater tanks at 26°C and under natural photoperiod. Animal experiments were performed following the regulations of the Guide for Care and approved by the Institutional Animal Care and Use Committee of Southwest University (No. IACUC-20181015-12, 15 October 2018).

### 4.2. Identification of RP genes from different chordates

The genomes of 13 species (sea squirt, lamprey, coelacanth, human, mouse, spotted gar, zebrafish, common carp, channel catfish, Atlantic salmon, medaka, tilapia and fugu) were examined to identify RP genes in each species. The genomic sequences of all species are available at the NCBI(https://www.ncbi.nlm.nih.gov/) and Ensembl (http://asia.ensembl.org/index.html) database. All RP genes were identified by tblastn (E = 2e^−5^) against genome sequences, using zebrafish and human RP proteins as the query sequences. The identified RP genes were named according to the principle described in the previous study [55]. Genomic distribution of RP genes was performed using NCBI and UCSC (http://genome.ucsc.edu/) databases.

### 4.3. Phylogenetic and Syntenic Analyses

Sequences were analyzed using the EditSeq and Megalign program with the Laser gene sequence analysis software package (DNAStar, Madison, WI, USA). Both amino acid and nucleotide sequences were used, and both maximum-likelihood (ML) and neighbor-joining (NJ) analyses were carried out for RP genes using MEGA 6.0 software (Tempe, AZ, USA) [56]. Phylogenetic trees for the small and large subunit RPs were generated by ML method with the amino acid sequences. Phylogenetic trees of the paralogous RP genes were constructed by NJ method with DNA sequences from a wide range of species. The degree of confidence assigned to nodes in trees was achieved by bootstrapping with 1000 replicates.

For syntenic analysis, position and orientation of RP genes and their adjacent genes on the chromosome were determined using Genomicus (available online: http://www.genomicus.biologie.ens.fr/genomicus-89.01/cgi-bin/search.pl) [53].

### 4.4. Expression Analyses of Tilapia RP Genes in Adult Tissues and Gonads at Different Developmental Stages

The transcriptomes of eight tissues from adult tilapia, including brain, heart, liver, ovary, testis, kidney, muscle and head kidney (Accession codes: PRJNA78915 and SRR1916191) [28], four pairs of XX and XY gonads from tilapia at 5, 30, 90, and 180 dah (Accession codes: SRA055700) [30] were downloaded from the NCBI database. Gonadal transcriptomes of control and secondary sex reversal (SSR) tilapia (Normal XX fish at 90 dah were fed a diet sprayed with 95% ethanol containing Fadrozole at a concentration of 200 μg/g diet for 90 days, while the control XX and XY fish were fed a diet sprayed with 95% ethanol but without Fadrozole) were reported by our previous study (Accession codes: SRP014017) [31]. Genome-wide expression analysis of RNA-Seq data was performed by TopHat2 v2.1.1 and Cufflinks [57]. RPKM was used to normalize the expression profile of RP genes. Only those transcripts mapped to unique loci in the tilapia genome were used to calculate RPKM values. Identification of XX/XY-enriched RP genes was performed as described previously [58,59]. Detection of genes differentially enriched was analyzed with Tbtools [60].

### 4.5. Validation of Differentially Expressed Genesby qRT-PCR and IHC

Two differentially expressed RP genes between ovaries and testes, RPL5b and RPL24, were selected to perform qRT-PCR analyses. The results were compared with the transcriptome data. To perform qRT-PCR, gonads were dissected from XX and XY tilapia at 5, 10, 20, 50, 70 and 180 dah, and total RNA was isolated from each sample and reverse-transcribed using MMLV reverse transcriptase (Invitrogen, Carlsbad, CA, U.S.A.) according to the manufacturer’s protocol. Gene-specific primers, RPL5b-F 5′- CCTGGTGCCTTCACGTGTTA-3′, RPL5b-R 5′- GTAGCCGGGGAAACGTTTC-3′, RPL24-F 5′- CCGATACGCCAGGATAGACG-3′, RPL24-R 5′- CCTTCTTGTGCTTGCGTCTG-3′, were used for qRT-PCR. qRT-PCR examination was performed according to the manufacturer’s protocol using the SYBR Green I Master Mix (TaKaRa, Dalian, China). Tilapia *eef1a1a* was used as internal control to normalize the expression of these two genes. The relative abundance ofmRNA transcripts was evaluated using the formula *R* = 2^−ΔΔCt^, as described previously [61]. Data are expressed as mean ± SD for triplicates. The statistical package GraphPad Prism (GraphPad Software, Inc. San Diego, CA, USA) was used to analyze data from qRT-PCR experiments. One-way ANOVA followed by posthoc test was performed to determine the significance. *p* < 0.05 was considered to be significantly different.

To validate which population of cells in the developing gonads expressed RPL5b and RPL24, IHC was performed using ovaries and testes from tilapia at 120 dah. Gonads were dissected, fixed in Bouin’s solution at room temperature overnight, embedded in paraffin and sectioned at 5 μm thickness. Paraffin sections were de-paraffinized and hydrated. The primary antibodies anti-mouse RPL5b(ARP56126_P050) and RPL24(ARP65377_P050) were purchased from AVIVA (Beijing AVIVA Systems Biology, Beijing, China). After overnight incubation with freshly diluted primary antibody (1000 times dilution) at 4 °C, the slides were washed twice in 1×PBS (phosphate-buffered saline) for 10 min, and then, incubated with anti-mouse IgG at room temperature for 1 h. After washing, 3,3′-diaminobenzidine tetrahydrochloride was applied for the color reaction, the slides were then counterstained in hematoxylin, dehydrated and mounted [62].

## 5. Conclusions

Comparative analyses of the RP genes in tilapia and other chordates provide a clear perspective on the evolution of RP genes. The number of the RP genes does not change much in vertebrates following 2R and 3R events, and significant expansion of RP genes was only observed in teleost fishes with 4R. The RP genes were found to be expressed in all tissues examined, but their expression levels differed among different tissues in tilapia, indicating their essential roles in various physiological processes. In addition, we also found, for the first time, that some of them displayed sexually dimorphic expression in developing gonads in fish. Taken together, these results present a new perspective to understand the evolution and function of RP genes in chordates and even other organisms. Future functional characterization of these RP genes with sexually dimorphic expression in teleosts will help us to better understand their important roles in sex differentiation and gonad development.

## Figures and Tables

**Figure 1 ijms-21-01230-f001:**
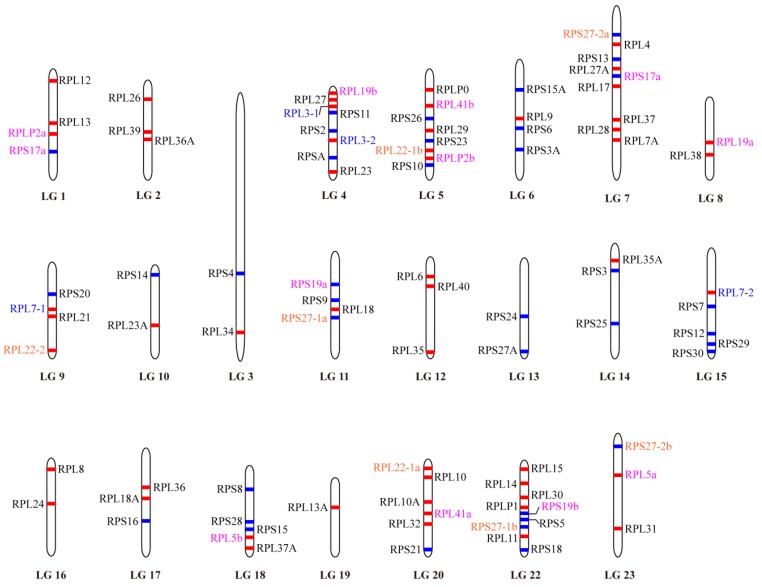
Chromosomal maps of the RP genes in Nile tilapia. RP genes are depicted on a linkage group (LG) of the genome (Release 2). LG21 is absent because it has been combined withLG16. Red and blue horizontal lines indicate the large subunit and small subunit RP genes, respectively. Gene name in blue indicates genes derived from 2R event. Gene name in magenta indicates genes derived from 3R event. Gene name in orange indicates genes derived from both 2R and 3R events.

**Figure 2 ijms-21-01230-f002:**
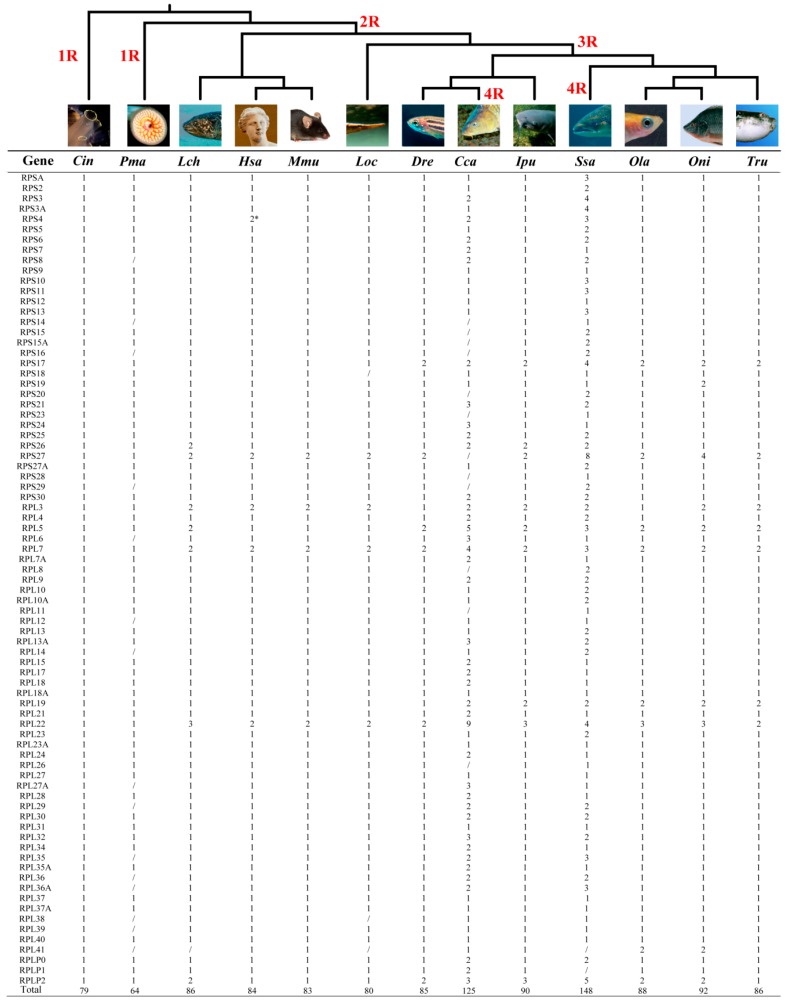
Phylogenetic relationship of 13 representative chordates analyzed and number variation of RP genes. 1R, 2R, 3R and 4R indicate the four rounds of WGD that occurred during vertebrate evolution. Slash indicates no homologous gene detected, probably due to secondary loss or incomplete genome sequences. * indicates two isoforms of RPS4 located on the human X and Y chromosomes. *Cin*, *Ciona intestinalis*; *Pma*, *Petromyzon marinus*; *Lch*, *Latimeria chalumnae*; *Hsa*, *Homo sapiens*; *Mmu*, *Mus musculus*; *Loc*, *Lepisosteus oculatus*; *Dre*, *Danio rerio*; *Cca*, *Cyprinus carpio*; *Ipu*, *Ictalurus punctatus*; *Ssa*, *Salmo salar*; *Ola*, *Oryzias latipes*; *Oni*, *Oreochromis niloticus*; *Tru, Takifugu rubripes*.

**Figure 3 ijms-21-01230-f003:**
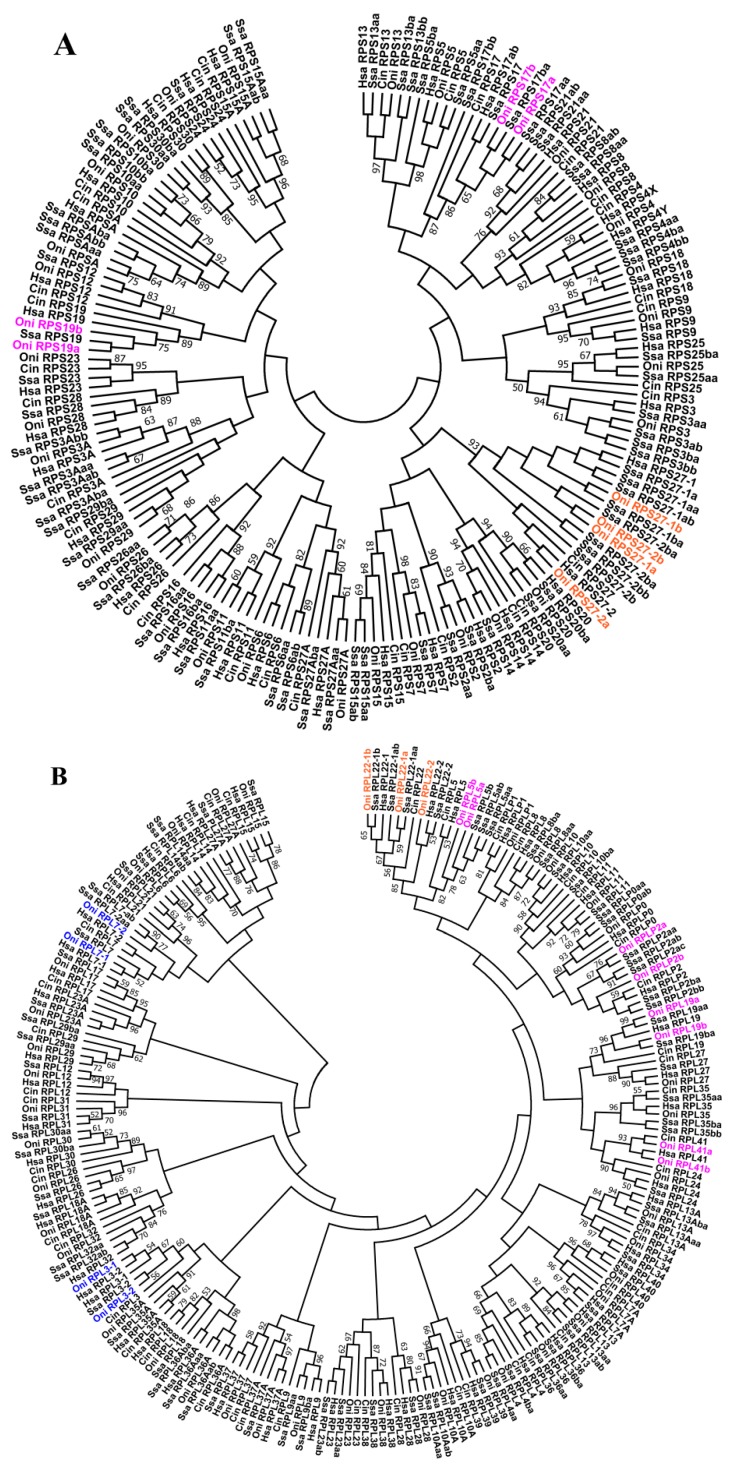
Phylogenetic tree of the small subunit (**A**) and large subunit (**B**) RP genes from sea squirt (1R), human (2R), tilapia (3R) and Atlantic salmon (4R). Gene name in blue indicates genes derived from 2R event in vertebrates. Gene name in magenta indicates genes derived from3R event in teleosts. Gene name in orange indicates genes derived from both 2R and 3R events. *Cin*, *Cionaintestinalis*; *Hsa*, *Homo sapiens*; *Oni*, *Oreochromis niloticus*; *Ssa*, *Salmo salar*.

**Figure 4 ijms-21-01230-f004:**
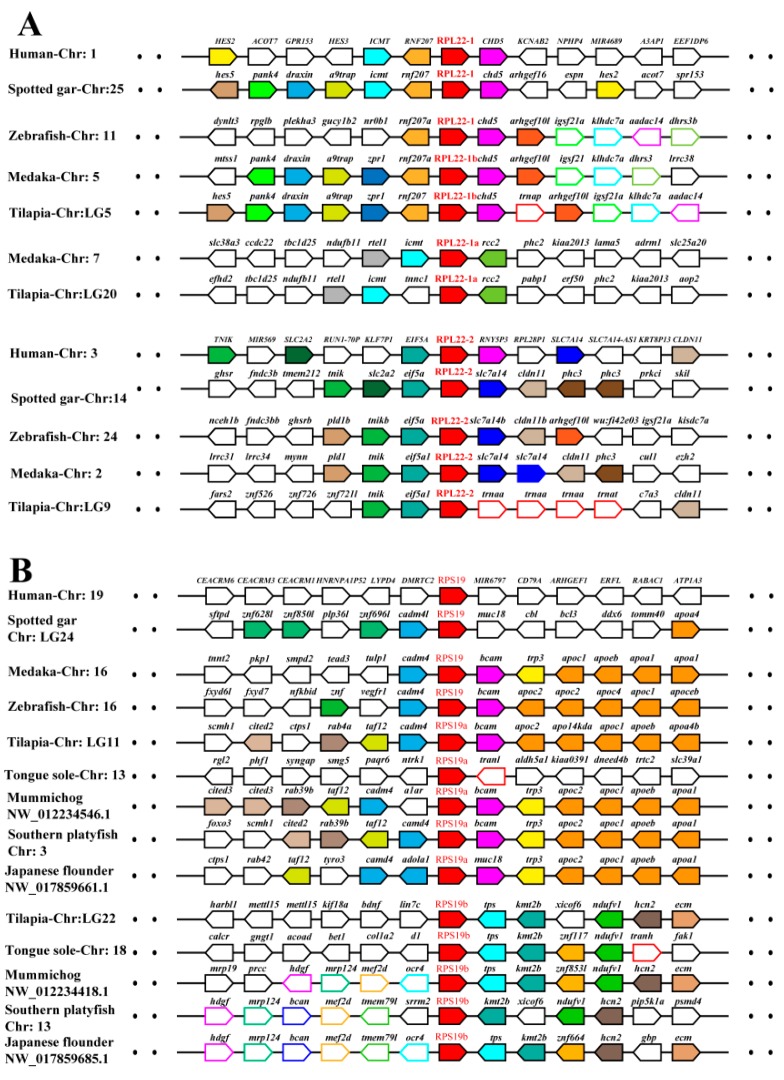
Synteny analyses of RPL22 (**A**) and RPS19 (**B**) and their adjacent genes in tilapia and other vertebrates. Rectangles represent genes in chromosome/scaffold. Dotted lines represent omitted genes of the chromosome/scaffold. The direction of the arrows indicates the gene orientation. The RP genes are shown in red, while the other genes are shown in different color.

**Figure 5 ijms-21-01230-f005:**
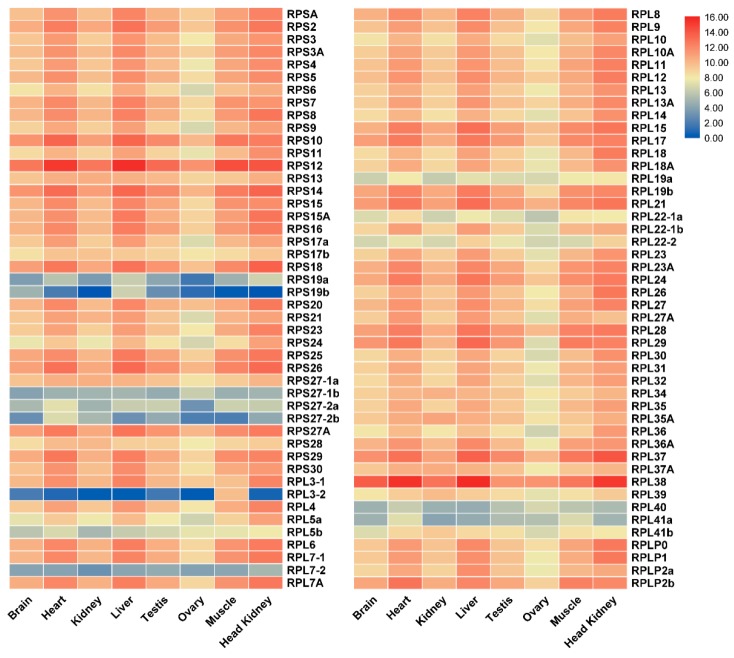
A heat map showing tissue distribution (RPKM: reads per kb per million reads) of RP genes in eight tissues based on transcriptome data in tilapia. Red and blue indicate high and low expression, respectively. Each row represents a different gene, and each column represents an independent tissue sample. The widespread complex expression patterns in all tissues were readily discernable. Most of the RP genes displayed high and ubiquitous expression in all tissues examined.

**Figure 6 ijms-21-01230-f006:**
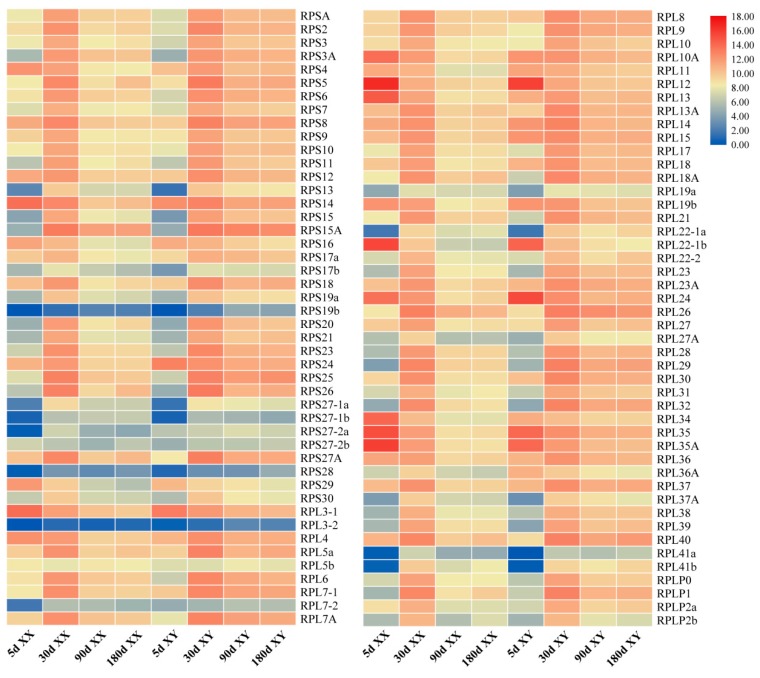
The expression profiles of RP genes in XX and XY gonads based on transcriptome data from tilapia. RNA preparations from gonads of XX and XY fish at 5, 30, 90 and 180 dah (day after hatching) were sequenced previously [30]. RPKM (reads per kb per million reads) was used to normalize the expression profiles of RP genes.

**Figure 7 ijms-21-01230-f007:**
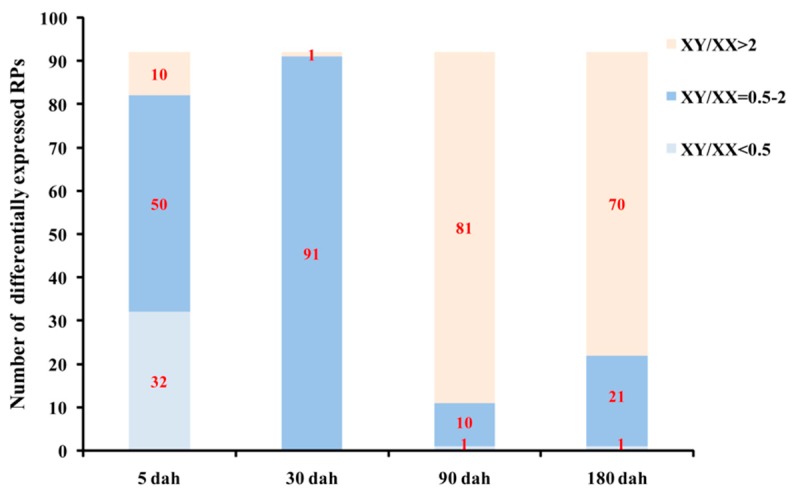
Number of differentially expressed RPs in XX and XY gonads of tilapia at 5, 30, 90 and 180 dah.XY/XX ratio was measured by RPKM values.

**Figure 8 ijms-21-01230-f008:**
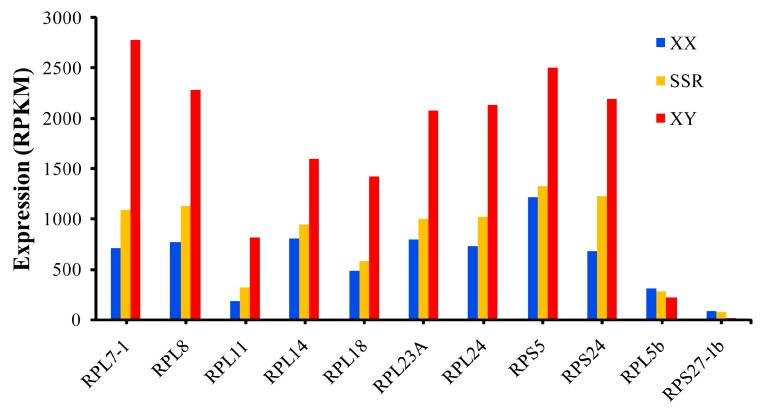
Sexually dimorphic expression of RP genes in XX and XY gonads based on gonadal transcriptome data from normal and secondary sex-reversal (SSR) tilapia [31]. SSR tilapia were obtained by feeding XX fish a diet sprayed with 95% ethanol containing Fadrozole at a concentration of 200 μg/g diet from 90 to 180 dah. The control XX and XY fish were fed a diet sprayed with 95% ethanol but without Fadrozole. RNA preparations from gonads of XX, XY and sex-reversed individuals at 180 dah were sequenced using Illumina 2000 HiSeq technology in our previous study. RPKM (reads per kb per million reads) was used to normalize the expression profiles of RP genes.

**Figure 9 ijms-21-01230-f009:**
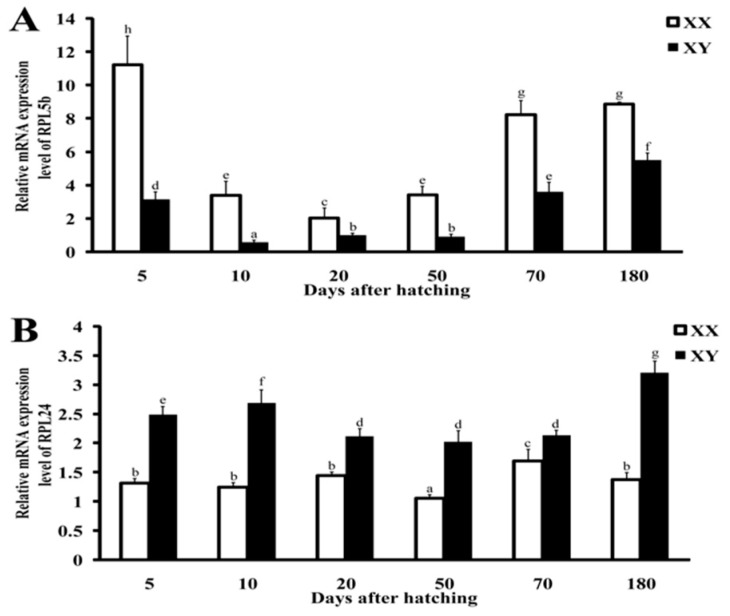
Ontogeny expression of RPL5b (**A**) and RPL24 (**B**) in tilapia gonads by qRT-PCR. Data are expressed as mean±SD for triplicates. Bars bearing different letters differ (*p* < 0.05) by one-way ANOVA followed by post-hoc test.

**Figure 10 ijms-21-01230-f010:**
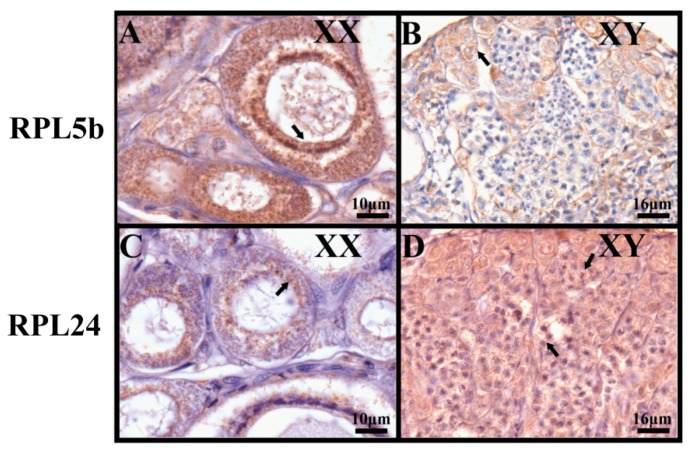
Sexually dimorphic expression of RPL5b and RPL24 in tilapia ovary and testis by immunohistochemistry. Samples were taken at 120 dah. Signals of RPL5b were observed mainly in the cytoplasm of oocytes in the ovary (**A**), while weak signals were detected in the spermatocytes in the testis (**B**). In contrast, very weak signals of RPL24 were observed in the cytoplasm of oocytes in the ovary (**C**), while strong signals were detected ubiquitously in different spermatogenic cells (**D**). Black arrows indicate positive signals. Scale bar, **A** and **C**, 10 μm; **B** and **D**, 16 μm.

**Table 1 ijms-21-01230-t001:** Ribosomal protein (RP) genes identified in Nile tilapia genome.

Gene	Accession Number	Chromosome Location	Exon Count	Gene Length (bp)	Protein Length (aa)
RPSA	LOC100707344	LG4	8	2836	307
RPS2	LOC100693457	LG4	7	2113	279
RPS3	LOC100696630	LG14	7	6031	245
RPS3A	LOC100690034	LG6	6	3940	266
RPS4	LOC100709648	LG3	7	7165	263
RPS5	LOC100691205	LG22	6	3187	203
RPS6	LOC100707655	LG6	6	4440	249
RPS7	LOC100704343	LG15	7	6283	194
RPS8	LOC100693356	LG18	6	4026	208
RPS9	LOC100694704	LG11	5	2450	194
RPS10	LOC100708197	LG5	6	4577	166
RPS11	LOC100696602	LG4	5	2026	161
RPS12	LOC100534443	LG15	5	2914	132
RPS13	LOC100696829	LG7	6	2934	151
RPS14	LOC100696222	LG10	5	2925	151
RPS15	LOC100695708	LG18	4	2627	145
RPS15A	LOC100698782	LG6	5	5166	130
RPS16	LOC100703910	LG17	6	3101	146
RPS17a	LOC100691912	LG7	5	7981	134
RPS17b	LOC100699764	LG1	5	5281	134
RPS18	LOC100699803	LG22	6	3445	152
RPS19a	LOC100689943	LG11	6	6770	146
RPS19b	LOC100690398	LG22	5	7095	152
RPS20	LOC100696588	LG9	4	1505	119
RPS21	LOC100701551	LG20	6	4172	83
RPS23	LOC100690746	LG5	4	2080	143
RPS24	LOC100700493	LG13	5	4576	131
RPS25	LOC100690179	LG14	5	2542	123
RPS26	LOC100700449	LG5	4	1749	115
RPS27-1a	LOC100705775	LG11	4	5189	84
RPS27-1b	LOC100696027	LG22	4	2376	84
RPS27-2a	LOC100691333	LG7	4	2298	84
RPS27-2b	LOC100690174	LG23	4	2477	84
RPS27A	LOC100697017	LG13	6	2770	156
RPS28	LOC100696499	LG18	4	2909	69
RPS29	LOC100703241	LG15	3	1056	56
RPS30	LOC100704053	LG15	5	1783	133
RPL3-1	LOC100711099	LG4	9	4294	403
RPL3-2	LOC100694819	LG4	10	3972	408
RPL4	LOC100694481	LG7	10	3520	369
RPL5a	LOC100708520	LG23	8	6819	297
RPL5b	LOC100700425	LG18	8	3792	298
RPL6	LOC100699139	LG12	6	6639	227
RPL7-1	LOC100702526	LG9	7	4215	245
RPL7-2	LOC100700318	LG15	7	4406	246
RPL7A	LOC100693537	LG7	8	3256	266
RPL8	LOC100706674	LG16	6	3269	257
RPL9	LOC100703191	LG6	8	5781	192
RPL10	LOC100697190	LG20	6	3989	215
RPL10A	LOC100702190	LG20	6	1982	216
RPL11	LOC100695881	LG22	6	3847	178
RPL12	LOC100695084	LG1	6	3525	165
RPL13	LOC100712089	LG1	6	4141	211
RPL13A	LOC100708460	LG19	7	3290	205
RPL14	LOC100534449	LG22	6	3561	137
RPL15	LOC100697988	LG22	4	2672	204
RPL17	LOC100707041	LG7	7	4249	184
RPL18	LOC100534549	LG11	7	5614	188
RPL18A	LOC100706131	LG17	5	3492	176
RPL19a	LOC100706003	LG8	6	2335	195
RPL19b	LOC100709589	LG4	6	3080	194
RPL21	LOC100711190	LG9	6	2393	160
RPL22-1a	LOC100699599	LG20	4	3986	129
RPL22-1b	LOC100710673	LG5	4	3264	129
RPL22-2	LOC100699889	LG9	4	2079	129
RPL23	LOC100701640	LG4	5	4091	140
RPL23A	LOC100705596	LG10	5	2677	155
RPL24	LOC100707648	LG16	6	3744	157
RPL26	LOC100708525	LG2	4	3651	145
RPL27	LOC100711912	LG4	5	3838	136
RPL27A	LOC100702758	LG7	5	6769	148
RPL28	LOC100704095	LG7	5	2011	137
RPL29	LOC106098935	LG5	4	3933	64
RPL30	LOC100710306	LG22	5	3112	116
RPL31	LOC100696251	LG23	5	3056	124
RPL32	LOC100534436	LG20	4	3255	135
RPL34	LOC109195715	LG3	5	2953	117
RPL35	LOC100711334	LG12	4	2608	123
RPL35A	LOC100711077	LG14	5	4038	110
RPL36	LOC 100691103	LG17	4	3201	105
RPL36A	LOC100690460	LG2	5	4870	106
RPL37	LOC100705081	LG7	4	4669	97
RPL37A	LOC100692490	LG18	4	2432	92
RPL38	LOC100711498	LG8	5	3295	70
RPL39	LOC100692256	LG2	3	3583	51
RPL40	LOC100699676	LG12	5	3438	128
RPL41a	LOC100711481	LG20	4	911	25
RPL41b	LOC100699101	LG5	4	1905	25
RPLP0	LOC100534569	LG5	8	2532	315
RPLP1	LOC100692069	LG22	4	2095	114
RPLP2a	LOC100691806	LG1	5	2962	114
RPLP2b	LOC100697185	LG5	5	1987	114

**Table 2 ijms-21-01230-t002:** Statistics of RP gene expressions in tilapia gonads at four developmental stages.

	5 dah		30 dah		90 dah		180 dah	
	XX	XY	XX	XY	XX	XY	XX	XY
Total	409,026	229,345	299,234	372,252	45,903	133,854	50,678	131,097
Average	4446	2493	3253	4046	499	1455	551	1425

Total indicates the total RPKM of all RP genes; average indicates the average RPKM of all RP genes.

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
