# Peer review of "Genome-Wide Identification, Evolution and Expression of the Complete Set of Cytoplasmic Ribosomal Protein Genes in Nile Tilapia"

_ijms, 2020, doi:10.3390/ijms21041230_

Round 1

Reviewer 1 Report

The manuscript “Genome-wide identification, evolution and expression of the complete set of cytoplasmic ribosomal protein genes in Nile tilapia” by Gangqian Kuang and collogues describes a computational identification and phylogenetic analyses of ribosomal protein genes in Oreochromis niloticus. With the plethora of genomic data available these days, we have entered an era where we can now perform high-throughput comparative and phylogenetic analyses of whole gene families to better understand how genomes and species have evolved. The Nile tilapia is an important aquaculture fish, thus understanding how its genome is organized and ultimately regulated, is important for understanding the basic biology of Nile tilapia. Overall, this manuscript describes a thorough comparative study of the ribosomal gene family. Several concerns were found and are summarized below.

Major concerns:

+ Figure 3 is missing.

+ The authors use data from various other species. Ideally, the accession numbers of the data used would facilitate clarity and reproducibility. The same is true for the accession numbers of the antibodies used.

+ Overall, the manuscript is rather verbose. Maybe the authors can improve the readability of the manuscript for clarity for the readers.

+ For clarity for the readers, ideally each section starts with a question the authors set out to answer. How the authors intend to address the question. Followed by the extensively described results.

+ The authors produced transcriptomics data, using the Illumina platform. It is common practice that the raw sequence data is made publicly available by providing the accessions numbers at any of the commonly used databases (NCBI, Ensembl, DDBJ, etc.).

Minor concerns:

+ The authors state on page 1, line 29, that “all cellular proteins required for growth.” Does this mean that protein production through ribosomes are not required during cellular maintenance (G0 or quiescence)?

+ Table 2 would benefit from an accompanied cladogram showing the major WGD events relative to the species studied in this manuscript.

+ Figure 2 should be larger, so it becomes easier to read.

+ The authors display the heat maps (Figures 4 and 5) for the expression levels of the various RP genes ranked according to their gene name. Would these heat maps be more understandable if these were ranked based on their expression levels? This reviewer thinks that especially Figure 5 will improve in highlighting the point the authors are trying to make. In addition, this reviewer thinks that in Figure 5, the x-axis should not follow the current comparison between XX and XY per dah time point, but instead a continuum of dah for first XX followed by XY. Alternatively, the authors might find it helpful if a heat map of differences in expression between XX (red) and XY (blue) is shown, as this is the ultimately the main point the authors report and discuss. This will also help the readers understand which genes the authors mention in the text (page 10, lines 11-13).

+ dah is not described in the main text. For clarity for the readers who are not experts in fish research, this should be done the first time this is mentioned (page 11, line 4).

+ The authors introduce sex-reversed fish without explaining what this is and how this is known to happen. Without a background, it is difficult for non-fish expert to understand and interpret Figure 7.

+ For the qRT-PCR and IHC the authors decided to focus on RPL5b and RPL24. Why did the authors focus on these two genes? Ideally, this reasoning is mentioned in the text.

+ Please try to make all the figures in the same font and with the same font-size where possible.

+ In the discussion, the authors mention that whole genome duplication events, followed by deletion or decay of some of the RP genes, as the major contributor to the diversity of mode of evolution of RP genes. For genes which are highly expressed, exist in many copies, and are essential for basic cellular functioning, the birth-and-death model of evolution is commonly inferred. Do the authors think this might be a contributing mode of evolution in the RP gene family?

+ On page 14-15 lines 50-2, the authors make a very bold statement that the differential expression of RP genes in the gonads “clearly triggered the biosynthesis machinery to adapt to the new challenging conditions”. In the absence of direct evidence, maybe it is more prudent to be a little less bold.

+ For the software used in this study, the authors should mention what settings they used, as this will allow others to reproduce the findings.

Reviewer 2 Report

The manuscript «Genome-wide identification, evolution and expression of the complete set of cytoplasmic ribosomal protein genes in Nile tilapia” submitted by Kuang and co-workers is a comprehensive study of cytoplasmic ribosomal genes in a teleost. The authors identified 92 RP genes in tilapia and studied both their expression patterns as well as their evolutionary origin.

For a researcher outside the RP field it is not obvious that these proteins are highly conserved even between vertebrates and invertebrates. This should be mentioned in the introduction. This sequence identity enables the authors to name their RPs with significant confidence and also enables the use of antibodies raised against mouse RPs on tilapia proteins.

In general, the manuscript lacks a lot of details to be reliable to the reader (see below) and needs a thorough rewriting and reanalyses to be published.  

The authors identify 92 RP genes in tilapia, where all seem functional. What about the number of pseudogenes as mentioned as an obstacle for identification of RP genes in other species? None in tilapia?

A short introduction to the nomenclature would also have been beneficial e.g. where RPL and RPS indicate large and small subunit.

The authors state the WGD origin of many RP genes but do not really show any phylogenetic evidence apart from phylogenetic trees including 5 tilapia RP genes in the main article and an additional seven tilapia genes included in trees shown in the supplementary. Without phylogenetic trees for the remaining sequences it is difficult for a reader or a reviewer to assess the WGD origin for the remaining 80 tilapia RP genes.

Minor issues:

Abstract: Line 17-18: “Transcriptome data of eight adult tissues showed that most of the RP genes expressed ubiquitously in all tissues, while their expression levels were highly variable among different tissues.”

Abstract: “There were remnants of both the 2R, the 3R as well as the 4R whole genome duplication events” but not in tilapia?

Page 2 Line 23: “proteinomics»?

Page 2 Line 28-33: “As is known to all, during fish gonadal differentiation and development, the male and female gonad each displays characteristic morphological and physiological features and patterns of gene expression. The gonadal model provides a unique opportunity to delineate the molecular mechanisms of the coordinated expression of RPs in the regulation of development. Thus, it is indispensible to identify the expression profiles of RP genes in the male and female tilapia.”

Page 2 Line 37-39: “….have made it an excellent model for genome-wide identification, tissue distribution and gonadal expression profile investigation of RP genes involved in sex determination and sex differentiation.” Are RP genes involved in sex determination?

Page 5 Line 23-27: “After comprehensive analysis, we identified 79, 84, 83, 84, 92, 152 and 125 RP genes in the lamprey, human, spotted gar, zebrafish, tilapia, Atlantic salmon and common carp genome, respectively. These results revealed that the number of the RP genes has undergone expansion in vertebrates following 2R, and significant expansion in teleost fishes due to 3R and 4R.” This statement is not correct and needs to be changed throughout the manuscript: 83-84 genes following 2R for human and spotted gar while 84 for zebrafish is not a substantial difference. Also, 79 RP genes were published in channel catfish, so unless they missed a lot (which can be checked in the available genome) the 1R to 2R to 3R is not highly supported. Also, the number of sequences does not correlate with 3R or 4R duplications without a phylogenetic analysis or a chromosomal analysis. For instance, Atlantic salmon also has a substantial number of genes that have duplicated after the 4R WGD so please provide a more detailed analysis of each gene to verify when the duplications occurred.

Page 5 Line 26-27: “Compared with Atlantic salmon, the absence of certain RPs in common carp may be due to incomplete genome sequencing and assembly.” There is a common crap genome available. Has this genome been investigated for presence of RP genes to verify the statement?

Page 13 Line 31-32: “We also updated the number of the RP genes from 79 to 84 in the zebrafish and human.” Where is this described and which genes were they?

Page 14 Line 14-16: “In the case of RPL22, the 3R duplication would explain the additional RPL22-1b gene observed in tilapia and other teleosts, which is closely phylogenetically related to other fish RPL22 sequences.”. Precision is lacking here: Who defined the RPL22-1a/b clade as two separate clades and which sequences belong to which a- or b- group? Based on previous nomenclature? Regardless, it is not obvious from the phylogeny that there is an additional RPL22-1b gene observed in teleosts as they are denoted -1, -1a, -1aa, -1ab or -1b. The legend should also include the abbreviations used for each species.

Page 14 Lines 16-17: “RPL5, RPL19 and RPS17 obtained a paralog during 3R duplication in fish.” A reference is needed to where the data showing this are presented? And the evolutionary relationship between RP5 sequences from various teleosts are more complex than all teleosts just having one additional gene originating from the 3R.

Page 14 Line 17-21: “Although the previous study indicated the RPS27 gene as a mammalian-specific isoform [45], the identification of four RPS27 genes in tilapia supports the hypothesis of two RPS27 paralogs in tetrapod and at least two in fish as a common feature. The two additional paralogous RPS27 genes might have appeared in the 3R or fish-specific genome duplication.” Add a reference to where these data are shown. And based on the data the two paralogs in tetrapods seem both to have duplicated in teleosts making four RPS27 genes more likely where a few species have lost some of these four.

Page 14 Line 21-22: “In summary, most of the duplicated RP genes from 2R were lost, but some of them, such as RPL3, RPL7, RPL22 and RPS27, were retained in vertebrates including human.” Where are the data showing the loss of 2R genes and where are the data showing the rest of the argument?

Page 14 Line 37-38: “Our work further supported the differences in the abundance of RPs and the expression profiles in tissues to be associated with distinct metabolic activity of tissues.” Where are the evidence for a link to which metabolic activities?

Page 15 Line 23: Reference to which NCBI and UCSC databases is missing.

Page 16 Line 33-35: “In addition, most RP genes were highly expressed in all eight tissues, while their expression levels were highly variable among the different tissues in tilapia, indicating their essential roles in various physiological processes”.

Material and Methods:

Phylogeny: Some of the trees have very low bootstrap values compared against the sequence supposed sequence identities. It is common to test both neighbour-joining but also maximum- likelihood trees to verify that the clustering is consistent. This should be done for trees where the clustering is not convincing.

Tables:

Table 1. “RP genes identified in Nile tilapia”. Include a description as to which genome and where it was found in the legend.

Figures:

The legend to figure 2 needs to include a reference to what the abbreviations are eg. Pma= Petromyzon marinus and Oni= Oreochromis niloticus etc. or a reference to another legend where the abbreviations are shown.

Figure 3 is missing

Figure 6: What is measured in the comparison XY/XX>2 etc.? RPKM values?

Figure 7: An explanation to SSR is needed.

Figure 9: Which day (dah) these samples were taken should be included in the figure legend.

Supplementary figures: Do the colors used in Figure S1 mean anything?

Round 2

Reviewer 2 Report

The authors have implemented the critisism and rewritten the manuscript accordingly. It is now acceptable for publication.
